# Preparation and Characterization of a Nano-Inclusion Complex of Quercetin with β-Cyclodextrin and Its Potential Activity on Cancer Cells

**DOI:** 10.3390/mi14071352

**Published:** 2023-06-30

**Authors:** Rajaram Rajamohan, Sekar Ashokkumar, Kuppusamy Murugavel, Yong Rok Lee

**Affiliations:** 1Organic Materials Synthesis Laboratory, School of Chemical Engineering, Yeungnam University, Gyeongsan 38541, Republic of Korea; 2Plasma Bioscience Research Center, Kwangwoon University, Seoul 01897, Republic of Korea; kumarebt@gmail.com; 3PG & Research Department of Chemistry, Government Arts College, Chidambaram 608 102, Tamil Nadu, India; ksmvel@gmail.com

**Keywords:** quercetin, β-cyclodextrin, nano-inclusion complex, nanoprecipitation process, feather-like images, MDA-MB-231 cells, fluorescence-assisted cell sorting analysis

## Abstract

Quercetin (QRC), a flavonoid found in foods and plants such as red wine, onions, green tea, apples, and berries, possesses remarkable anti-inflammatory and antioxidant properties. These properties make it effective in combating cancer cells, reducing inflammation, protecting against heart disease, and regulating blood sugar levels. To enhance the potential of inclusion complexes (ICs) containing β-cyclodextrin (β-CD) in cancer therapy, they were transformed into nano-inclusion complexes (NICs). In this research, NICs were synthesized using ethanol as a reducing agent in the nanoprecipitation process. By employing FT-IR analysis, it was observed that hydrogen bonds were formed between QRC and β-CD. Moreover, the IC molecules formed NICs through the aggregation facilitated by intermolecular hydrogen bonds. Proton NMR results further confirmed the occurrence of proton shielding and deshielding subsequent to the formation of NICs. The introduction of β-CDs led to the development of a distinctive feather-like structure within the NICs. The particle sizes were consistently measured around 200 nm, and both SAED and XRD patterns indicated the absence of crystalline NICs, providing supporting evidence. Through cytotoxicity and fluorescence-assisted cell-sorting analysis, the synthesized NICs showed no significant damage in the cell line of MCF-7. In comparison to QRC alone, the presence of high concentrations of NICs exhibited a lesser degree of toxicity in normal human lung fibroblast MRC-5 cells. Moreover, the individual and combined administration of both low and high concentrations of NICs effectively suppressed the growth of cancer cells (MDA-MB-231). The solubility improvement resulting from the formation of QRC-NICs with β-CD enhanced the percentage of cell survival for MCF-7 cell types.

## 1. Introduction

Natural food additives known as polyphenols were present in a wide variety of plant foods, including strawberries, apples, onions, grapes, tea, and coffee [1,2,3]. Numerous studies have demonstrated that polyphenols have a wide range of biological effects, including immunological support, cardiovascular protection, anticancer, antibacterial, anti-inflammatory, and antioxidant properties [4,5,6,7,8,9]. In recent discoveries, researchers have identified that plant extracts possess cytotoxic properties, making them potential candidates for various applications [10,11,12,13,14]. Due to their benign qualities and reduced adverse effects on the body, plant-derived chemopreventive and chemotherapeutic drugs were currently receiving more and more attention on a global scale [15,16]. The primary dietary polyphenol, quercetin, has been recognized for its in-vivo and in-vitro anticancer properties against liver, breast, prostate, and colon cancers. Su and colleagues discovered that the expression levels of p-AMPK and p-p70s6k were increased and downregulated, respectively, when bladder cancer cells (T24, UMUC3, and MB49) were treated with quercetin, demonstrating that quercetin induced cell death via activation of the AMPK signaling pathway [17,18].

Despite QRC’s widespread use as a dietary supplement, its low bioavailability severely restricts its potential for use in medicinal applications. Inclusion complexes between QRC and CDs have been described in several research reports as a means of improving QRC’s solubility [19,20,21]. CD-based QRC has been the subject of numerous modification studies for both biological and delivery systems. Recently, solid complexes were prepared and employed for biological applications using QRC with β-CD and also modified CDs [22,23,24,25,26]. For instance, Zeynep et al. constructed nanofibers between QRC and β-CD in order to accomplish slow-release behavior [27]. With the same inclusion complexes, the Nutsarun research group created a hydrogel-like substance for wound healing [28]. Recently, our research team developed MOFs for use in antibacterial and antifungal applications based on supramolecular assembly between QRC and β-CD [29].

Some techniques were available for obtaining materials with nanoscale supramolecular assembly for transport and biological applications [30,31,32]. However, the majority of these techniques concentrate on chemical techniques that significantly alter their structure and properties. Additionally, several hazardous chemicals were produced during the process, which may compromise the quality of food and pharmaceutical items [32]. Therefore, future research must take into account secure, eco-friendly, and sustainable techniques. A quick, repeatable, and easy method for creating CD-based nanocomposites was nanoprecipitation [33,34]. Precipitation entails the continual addition of non-solvents to CD IC solutions that have already been produced, or vice versa [35,36]. The non-solvents were typically non-toxic. The quick production of nanoparticles was caused by turbulence at the interface between the solvent and non-solvent.

Pharmaceuticals for antibacterial and anticancer medication delivery, healthcare for tumor detection, and agriculture for plant protection and nourishment were just a few of the domains where nanotechnology has been a game-changer [37,38,39,40,41]. The term “nanomedicine” refers to a field of medicine that makes use of nanotechnology platforms and enables researchers and clinicians to create intelligent nanoscale carriers, typically between 20 and 150 nm in size. Scientists may now create intelligent nanoscale carriers for nanomedicine thanks to nanotechnology. These carriers overcome different mechanical and physiological obstacles while carrying therapeutic or diagnostic molecules, resulting in the best possible drug delivery and diagnostic results. As of now, a number of multifunctional working nanocarriers have been formed that were mainly based on the supramolecular self-assembling of various molecules, with lipids serving as the most prominent example. Cyclodextrins (CDs) have received attention over the past few decades as potential building blocks for various nanoparticles [42,43,44]. The macrocyclic oligosaccharides known as cyclodextrins were composed of glycosyl units connected by (1→4) glycosidic linkages. Among them were α, β, and γ-CDs consisting of 6, 7, and 8 glucosyl units, respectively, which were produced by the enzyme cyclodextrin glycosyl transferase through the enzymatic conversion of starch [45,46]. By trapping a range of hydrophobic molecules in their apolar chamber, CDs and their hydrophilic derivatives create inclusion compounds that improve drug solubility, drug stability, absorption, and permeability across biological barriers [47,48]. There were certain inclusion complex products on the market that were meant to be administered by oral, parenteral, nasal, rectal, and ocular routes, attesting to the biocompatibility and safety of these CDs and their hydrophilic derivatives [49,50,51].

Additionally, according to Han et al. (2022), [52] nanoparticles produced by the nanoprecipitation process typically have greater physicochemical stability and bioavailability. In conclusion, there were a lot of benefits to using the nanoprecipitation process. However, we have found no reports on the nanoprecipitation-based production of QRC:β-CD NICs in the literature. This work was partially inspired by the fact that there was still a big knowledge gap regarding the physicochemical characteristics, bioavailability, and stability of QRC:β-CD NICs. This work investigated the creation of QRC:β-CD NICs to enhance QRC delivery in order to address the issues of low bioavailability and poor stability of QRC. Furthermore, the utilization of a nanoprecipitation method to create QRC:β-CD NICs was crucial to extending the use of QRC in anticancer effectiveness.

## 2. Materials and Methods

### 2.1. Materials

Sigma-Aldrich Chemical Co., Ltd. (Daegu, Republic of Korea) provided the quercetin (C_15_H_10_O_7_, Molecular Weight: 302.23, Purity > 95%) and β-CD (C_42_H_70_O_35_, Molecular Weight: 1134.98, Purity > 97%). Additionally, Sigma-Aldrich Chemical Co. (Daegu, Republic of Korea) provided absolute ethanol. Other compounds were all of the analytical grades.

### 2.2. Preparation of QRC:β-CD NICs

The following method was used to create QRC:β-CD NICs [33]. The 1:1 molar ratio was used for the preparation of NICs between QRC and β-CD, according to the report of QRC:β-CD-ICs [25]. First, an aqueous solution of β-CD (10 mM) was made by dissolving β-CD (67.1 mg) in water (10 mL). Then, while continuously stirring, 2 mL of 100% ethanol (absolute ethanol), containing 22.7 mg of QRC, was added dropwise using a syringe. The mixes were then vigorously agitated for five hours while kept in the dark (Figure 1). Then, using the nanoprecipitation technique, QRC:β-CD NICs were synthesized [52]. Through a syringe pump, about 66.0 mL of absolute ethanol was utilized as a non-solvent (reagent) and was added in a dropwise manner to QRC:β-CD-ICs solution while being stirred. The mixes were then subjected to 20 min of 200 rpm magnetic stirring. The sediment of NICs, which had spontaneously formed, was separated by centrifugation (20 min), freeze-dried, and kept at 4 °C for further use.

### 2.3. In-Vitro Cytotoxicity Assay

The details of the Materials and Methods of in-vitro cytotoxic assay and Apoptosis study were provided in the Appendix A.

### 2.4. Statistical Analysis

Statistical analysis was conducted utilizing Microsoft Excel 2010. The significance of the treatments was evaluated through Student’s *t*-test. A *p*-value below 0.05 was considered to indicate statistical significance. The data were presented as means accompanied by standard errors.

## 3. Results and Discussion

### 3.1. Surface Characterization of QRC:β-CD NICs

#### 3.1.1. Imaging of NICs by FE-SEM Analysis

The obtained NICs’ physical appearance revealed a yellowish color that was slightly paler than the color of QRC in its raw form. The NICs appeared feather-like when self-aggregating, as demonstrated by the FE-SEM images. The feather-like structures can be seen with a 2 μm scale at higher magnification (Figure 1A–D). It was determined that the typical particle size was 250 nm. The fact that QRC:β-CD-NICs have bigger particle sizes suggests that NIC size increased as the system β-CD content increased [52,53,54]. It was generally accepted that the nucleation of a small collection of molecules, followed by the aggregation of these nuclei, was what causes nanoparticles to form via nanoprecipitation [52]. The likelihood of particle collision, resulting in the formation of larger agglomerated particles, increases with a higher concentration of β-CD. This was due to the increased number of particles generated per unit volume in the solution during the nanoprecipitation process. As a result, the size of the QRC:β-CD NICs obtained at high β-CD concentrations is bigger. The synthesized QRC:β-CD NICs were arranged as feathers on a regular basis. The NICs were well distributed, but the SEM pictures showed that some of them were grouped together to form bigger agglomerates, which was explained by the hydrogen bonding connections between the NICs. QRC:β-CD ICs were micron-sized structures, according to earlier investigations [29]. In contrast to the micron-sized QRC:β-CD ICs synthesized through co-precipitation, the microstructure of the QRC:β-CD NICs produced in this study exhibited notable differences. The distinct morphological characteristics observed further confirmed the successful synthesis of QRC:β-CD NICs via the nanoprecipitation process.

#### 3.1.2. Imaging of NICs by AFM Analysis

AFM utilizes a nanoscale probe to scan the NIC sample, generating precise topographic images. This enables the analysis of surface texture and roughness at the nanoscale, even when the size of the surface feature was significantly smaller than 3 μm [55]. Figure 1E–J depicts the 2D and 3D pictures revealing a minor aggregation of NICs formed between QRC and β-CD. The surface morphologies of the NICs were comparable because of the homogenous particle distribution in the area of view. Strong evidence that the NICs were generated was provided by changes in the surface topography and crystal structure of the particles throughout time [56,57]. The NICs’ surface roughness was a little better, and this further verified the inclusion of the QRC and β-CD. According to the results of the AFM measurements, the sample contained isolated complexes that were stable and did not undergo a structural change when scanning with the AFM tip [58].

#### 3.1.3. Imaging of NICs by HR-TEM Analysis

Transmission electron microscopy (TEM) was utilized to confirm the size and size distribution of the NICs in a dry state prior to performing DLS analysis. Figure 1K–Q displays the TEM micrographs of the NICs. The NICs were evenly dispersed across the entire surface in complete aggregation and were weakly connected to one another. The micrographs support the SEM pictures by demonstrating the feather-like structure that had been identified. Each feather structure was about 40 nm in size. We may presume that the core diameter derived from TEM images was fairly near to the diameter of the β-CD and QRC because they make up an equal portion of the NICs. The amorphous nature of the NICs and the absence of a crystalline phase were verified through the analysis of the selected area electron diffraction (SAED) pattern (Figure 1R). The generated NICs had no chance of containing the crystalline indexes. Furthermore, the TEM data could potentially provide insight into the crystalline or amorphous characteristics of the generated NICs, as inferred from the XRD pattern.

#### 3.1.4. Dynamic Light Scattering (DLS) Analysis

Using the dynamic light scattering (DLS) technique, which exposes the average hydrodynamic diameter of particles in a liquid suspension, further characterization was performed to identify the particle size distributions in the NICs. The intensity particle size distribution measured for the NICs was shown in Figure 2A. The graphic contrast various size classes with the relative percentage of light scattering by the particle. As can be seen in Figure 2A, the distribution curve of NICs displayed a single peak that was situated between 200 and 400 nm. The NPs could form a uniform suspension because of their good dispersion in solution and average particle size of 206.7 ± 4.15 nm and PDI of less than 0.8 [34,59]. However, for particle sizes that grow for diameters greater than 25 nm, the bandwidth increases with the increase in particle size [60,61]. The study of NICs revealed no further particles of any other size, with the intensity percentage for the particle size being determined to be 100 (Appendix A).

#### 3.1.5. XRD Analysis

In Figure 2B, XRD patterns for the QRC, and NICs were shown. The crystalline nature of β-CD was demonstrated by a number of diffraction peaks of β-CD with 2θ positions at 8.98, 10.64, 12.46, 15.38, 17.66, 18.78, 22.68, 25.62, and 34.72° [52,62]. This β-CD pattern was in line with the packing of the molecules in the molecular structure that resembles a cage. The crystalline structure of QRC was clearly demonstrated by strong peaks in its XRD patterns at 9.21, 12.25, 13.81, 16.21, 26.78, and 28.15°. The NICs were successfully synthesized and had a novel phase of crystallinity, as shown by the markedly different XRD patterns of QRC:β-CD NICs and QRC. This was additional circumstantial proof of morphological modifications in NICs. In addition, after the formation of NICs, the strength of some QRC diffraction peaks at 12.25 and 26.78° was reduced. A few of the peaks of β-CD were also merged between 15 and 22°, indicating the presence of β-CD in the NICs. The disappearance of the peaks at positions 15.38, 17.66, and 18.78° in the NICs suggests that QRC was successfully encapsulated in an amorphous state. Additionally, the generated NICs were amorphous in nature rather than crystalline, as seen by the XRD pattern that was reflected in the TEM results. According to Chen et al. (2021), bioactive substances with amorphous structures have substantially better bioavailability than those with crystalline forms. As a result, it may be assumed that QRC’s bioavailability in NICs might be improved [63].

#### 3.1.6. Thermal Analysis (DSC Analysis)

The DSC curves of QRC and QRC:β-CD NICs were shown in Figure 2C. The elimination of water molecules from the QRC was indicated by a broadened endothermic peak at 144.22 °C [64]. The spectral measurement ranges extend from 40 to 250 °C. Within these ranges, no additional endothermic peaks have manifested. Contrarily, β-CD displayed a large endothermic peak at 113.95 °C, which was associated with the dehydration of β-CD [33,65]. Similar to this, the NICs also produced the same endothermic peak for moisture removal (removal of water molecules) at 142.25 °C. The initial β-CD hydrate and the NICs exhibited comparable behavior, with the exception of the anhydrous β-CD transitioning from a crystalline to an amorphous form at 213.73 °C [66]. This abrupt endothermal effect, which occurs at 213.73 °C, was observed in the NICs, confirming the occurrence of the molecular inclusion process and the subsequent breakdown of the β-CD crystal structure [67,68]. The amorphous condition of QRC caused by nanoencapsulation was confirmed by the above results, which were in good accord with the XRD findings.

### 3.2. Chemical Characterization of QRC:β-CD NICs

#### 3.2.1. FT-IR Spectral Analysis

Figure 3A shows the FT-IR spectra of QRC and QRC:β-CD-NICs. It was evident that the wider O-H bond for the characteristic peak of the β-CD was situated at around 3300 cm^−1^ [65,69]. While the band at 1650 cm^−1^ resulted from polysaccharide C-C stretching, the band at 2930 cm^−1^ was assigned to the asymmetrical stretching vibration of C-H bonds. The C-H bending vibrations were attributed to the reflections from the hyperfine structure in the range of 1480 cm^−1^ to 1180 cm^−1^. The band between 1200–1000 cm^−1^ was indicative of the symmetrical stretching of glycosidic C-O, and C-O-C bonds in the polysaccharides [70]. Pure QRC’s FT-IR spectrum was depicted in Figure 3A, where its distinctive bands [71,72] were found. While OH bending of the phenol function was apparent at 1379 cm^−1^, OH group stretching was discernible between 3406 and 3283 cm^−1^. At 1666 cm^−1^, the C=O aryl ketonic stretch absorption was evident. At 1610, 1560, and 1510 cm^−1^, extending bands of the C=C aromatic ring can be seen. C-H in aromatic hydrocarbons showed an in-plane bending band at 1317 cm^−1^, and out-of-plane bending bands can be seen at 933, 820, 679, and 600 cm^−1^. The bands at 1263, 1200, and 1165 cm^−1^ can be attributed to the C-O stretching in the aryl ether ring, the C-O stretching in phenol, and the C-CO-C stretching and bending in ketone, respectively. The FT-IR spectral analysis reveals that due to the low content of QRC in QRC:β-CD NICs and the overlapping bands of QRC and β-CD, there was no distinct difference observed between the FT-IR spectra of QRC, β-CD, and NICs [71,73]. This observation suggests that QRC was successfully encapsulated within the β-CD cavity in NICs, resulting in certain QRC peaks being less pronounced or not clearly visible in NICs. Furthermore, the FT-IR spectra of NICs containing both QRC and β-CD exhibit slight shifts in the peaks of β-CD, specifically at 3300 cm^−1^, and 2930 cm^−1^. Additionally, the NICs’ O-H, C-H (asymmetric), and C-C vibrations were represented by the peaks that emerge at 3306, 2922, and 1656 cm^−1^, respectively. The peaks stated above showed that the collected NICs contained QRC and β-CD. Additionally, the peaks for QRC for the vibrations of O-H, C=O, C-O (stretched), and C-O (phenolic) were slightly different and were connected to the formation of NICs by hydrogen bonding between the QRC and β-CD, as indicated in Appendix A. The following were the causes of this interaction: QRC was stabilized within the cavity of β-CD via the intermolecular hydrogen bonding interactions, (i) when one or both of its aromatic rings entered the cavity and caused the strong hydrogen bonding [74], and (ii) when the IC molecules clumped together to form NICs.

#### 3.2.2. Raman Analysis

Figure 3B shows the Raman spectra of the QRC and NICs. Peaks in the QRC can be found at location 1627.37 cm^−1^, which was related to the C=O stretching. The O-H bending was also responsible for the peaks at coordinates 1564.59, 1437.59, and 1326.64 cm^−1^ [75,76]. The peak position was at 1114.96 cm^−1^ for the C-C stretching in the NICs [77]. The above-mentioned peak locations in the NICs’ Raman spectra were 1614.96, 1554.74, 1453.64, 1336.86, and 1127.00 cm^−1^, respectively. Position changes after the addition of β-CD show how the inclusion complexes had taken shape and their formation. The Raman spectrum shifts took place regardless of whether they formed NICs.

#### 3.2.3. NMR Analysis

In the ^1^H NMR spectra, the methine and methylene protons of the β-CD ring display multiplet signals at 3.30, 3.56, and 3.63 ppm (Appendix A). The presence of a triplet signal at 4.45 ppm, with a coupling constant of approximately 12 Hz, can be attributed to the methine proton coupled to the hydroxy methylene group. Two signals appearing as a doublet at 4.82 and 5.67 ppm were assigned to the hydroxy group located in the equatorial position of the β-CD ring. The proton signal observed at 5.73 ppm, with a coupling constant of 6.6 Hz, corresponds to the equatorial hydrogen of the β-CD. Additionally, the multiplet detected at 3.33 ppm can be assigned to the hydroxy proton connected to the methylene group.

Five proton integrals caused by aromatic protons were observed in the region of 6.18–7.67 ppm in the ^1^H NMR spectra of free QRC (Figure 4A). Based on the coupling constant values and its substituent effect, the individual aromatic proton and hydroxy group assignments for QRC were determined. The aromatic protons of the benzene ring that fused with oxopyran were believed to be responsible for the doublets seen at 6.18 and 6.40 ppm with a coupling constant of 1.8 Hz. The doublet observed at 6.88 ppm, with a coupling constant of 8.4 Hz, was attributed to the protons of the phenyl group connected to the bezoxopyran ring. Additionally, the doublet of a doublet detected at 7.53 ppm, with coupling constants of 8.4 and 2.4 Hz, and another doublet with a modest coupling constant of 2.4 Hz observed at 7.67 ppm, can also be attributed to the protons of the phenyl group linked to the bezoxopyran ring. The protons found at 6.88 and 7.53 ppm were close to one another based on the coupling constant values. Broad signals were observed for the hydroxy protons in the benzene ring at 9.35 (2H), 9.60 (1H), 10.80 (1H), and 12.48 ppm (1H). The QRC proton chemical shifts were consolidated and provided in Appendix A.

Through examination of the NMR spectral patterns of the NICs, a complexation was proven. All of the proton signals resulting from QRC and β-CD were detected with identical intensities in the NMR spectrum. Additionally, correlation spectroscopy verified complexation. Most protons in the complex’s NMR spectra had deshielded by 0.03–0.06 ppm. These protons were protected by 0.16 ppm, with the exception of the hydroxyl protons of the benzopyran ring (Figure 4B). The signals were individually assigned by contrasting the proton chemical shifts of QRC and β-CD. The singlet at 12.54 ppm in the proton NMR spectrum was identified as a hydroxy proton at C-2. The four hydroxy protons of the phenyl rings were attributed to the broad signals seen at 10.68 and 9.42 ppm. The aromatic protons of the benzopyran ring were responsible for the proton chemical shifts at 6.23 and 6.45 ppm, while the signals at 6.93, 7.60, and 7.73 ppm were conveniently attributed to aromatic protons of the phenyl ring connected to the pyran ring. At 3.33, 3.62, and 3.70 ppm, the methine protons of the β-CD were seen as multiplets. At 4.51 ppm, a triplet of another axial methine proton with a CH_2_OH substituent was detected. The methine proton observed at 3.70 ppm was combined with the proton signal caused by the methylene group. The equatorial hydroxy protons and one equatorial hydrogen of the β-CD were readily ascribed to the proton signals found at 4.89 and 5.75 ppm. The substituent effect was used to make the individual assignments of the proton signals, which were shown in Appendix A.

##### ROESY Analysis of QRC:β-CD NICs

ROESY provided additional confirmation of complex formation (Figure 5A). In the ROESY, there was a correlation between the cross peak at 3.33 ppm and 9.42, 7.73, 5.75, 4.89, and 4.51 ppm. The methine proton-bearing hydroxy methylene group has already been given credit for the proton chemical shift at 4.51 ppm. The equatorial hydroxy protons of the CD ring were responsible for chemical shifts at 4.89 and 5.75 ppm. The remaining proton chemical shifts were caused by hydroxy protons at C-3′ and C-4′ and H-2′ protons at 7.73 and 9.42 ppm, respectively. Consequently, one of the methine protons of the β-CD component interacts with both the aromatic proton of the connected phenyl ring to the benzopyran ring and the hydroxy protons. On the basis of the aforementioned correlations, a suggested structure was as below (Figure 5B).

### 3.3. Cytotoxic Assay on MCF-7, and MDA MB 231 Cancer Cell Lines

The cytotoxicity assays were conducted using different cancer cell lines. The QRC and NICs were introduced to the cells, and their effects were measured through various parameters such as cell viability, and apoptosis. The experiments were designed to compare the cytotoxic potential of the NICs against that of the active compound alone, as well as against a control. The in-vitro cell viability was determined by performing the Alamar blue assay on MRC-5 and MDA-MB-231 cell lines [41]. The results, expressed as mean  ±  SD (n = 3), were presented in Figure 6. In the case of the MRC-5 cell line, the survival of cells treated with both QRC and its NICs was carefully monitored and quantified (Figure 6A). It was observed that concentrations up to 20.0 μg/mL had no discernible toxic effect on the cells. However, higher concentrations of 200.0 μg/mL and 2000.0 μg/mL exhibited a relatively less toxic effect after a 3-day treatment period. On the other hand, when assessing the MDA-MB-231 cell line, the survival of cells showed a gradual decrease by the third day of QRC treatment. These cells demonstrated resilience up to concentrations of 20 μg/mL, but beyond that threshold, a significant impact on cell survival was observed (as depicted in Figure 6B). The differential responses observed between the two cell lines suggest that MRC-5 cells display a higher tolerance to QRC and its NICs compared to the MDA-MB-231 cells (Appendix A). Compared to QRC alone, the presence of high concentrations of QRC NICs inclusions demonstrated a reduced toxic effect on normal human lung fibroblast MRC-5 cells. Furthermore, both low and high concentrations of QRC NICs, whether administered separately or in combination, effectively inhibited the growth of cancer cells (MDA-MB-231). These findings highlight the importance of considering the specific cell type when evaluating cytotoxicity and potential therapeutic effects of QRC and its NICs. Further investigations were necessary to elucidate the underlying mechanisms and explore the potential application of QRC and its NICs in targeted cancer therapy.

The bright-field fluorescence microscopic images provided valuable insights into the morphological changes of the cell lines under investigation. In Figure 7A, it can be observed that the control group displayed no noticeable morphological alterations. Similarly, when examining the cells treated with both QRC and NICs for both the MRC-5 and MDA-MB-231 cell lines, no significant morphological changes were observed. To further assess the in vitro toxicity of MDA-MB-231 cells, an apoptosis assay was conducted using a flow cytometer (Figure 7B). A concentration of 1.0 mg/mL of QRC and NICs were used for this analysis. The assay employed combined markers, Annexin V and PI, which were indicative of cell apoptosis and necrosis. Remarkably, the results demonstrated that there were no significant levels of cell apoptosis or necrosis observed in comparison to the control group. The percentages of apoptosis observed in different conditions were as follows. For the control measurement, Lower Left (LL): 98.1%, Upper Right (UR): 0.5%, and Lower Right (LR): 1.5%. For the QRC material, LL: 81.4%, UR: 7%, LR: 11.6%, and for the NICs, LL: 86.7%, UR: 5.7%, LR: 7.6%. Taken together, the findings from the bright-field fluorescence microscopy analysis and the apoptosis assay using flow cytometry indicated that treatment with QRC and NICs did not induce substantial morphological changes or significant levels of cell apoptosis or necrosis in the MRC-5 and MDA-MB-231 cell lines. These results contribute to a better understanding of the safety profile and potential non-toxic nature of QRC and its NICs in the context of these cell lines.

The mechanism of cytotoxicity with QRC and NICs can involve the pathway of enhanced cellular uptake. NICs can improve the solubility and stability of cytotoxic agents, leading to enhanced cellular uptake. This increased intracellular concentration of the active compound can promote its interaction with intracellular targets, resulting in cytotoxic effects [78].

## 4. Limitations of the Study

Overall, the findings of this study support the potential of NICs as effective cytotoxic agents against cancer cells. The ability of CD to encapsulate and deliver anticancer compounds could be harnessed to improve the therapeutic outcomes of existing drugs and potentially overcome drug resistance in cancer treatment. Further research and in-vivo studies were warranted to validate the efficacy and safety of these cyclodextrin complexes for clinical applications.

## 5. Conclusions

In this study, we have successfully synthesized QRC:β-CD NICs utilizing a nanoprecipitation method that was simple, flexible, and efficient. The NICs exhibited a compact particle size, measuring approximately 200 nm. Through various analyses, such as SEM, DSC, XRD, FT-IR, and NMR, the formation of NICs was confirmed. FT-IR analysis demonstrated the presence of hydrogen bonds between QRC and β-CD, while intermolecular hydrogen bonding facilitated the aggregation of IC molecules to form NICs. Proton NMR results provided clear evidence of proton shielding and deshielding following NIC formation. The addition of β-CDs resulted in the emergence of a feather-like structure in the NICs. The particle sizes were consistently around 200 nm, and there was no evidence of crystalline NICs as supported by SAED and XRD patterns. We conducted in-vitro cytotoxicity testing on cell lines, MCF-7 and MDA MB 231, to assess the potential of QRC:β-CD NICs as effective oral formulations of QRC. The results indicated that these NICs could serve as a promising strategy for developing highly efficient oral formulations of QRC. However, it was imperative to prioritize safety and regulatory approval before considering their application in foods, supplements, and medicines. Looking ahead, there were promising prospects for nano-sized CD delivery systems in the biological and food industries, including applications in active food packaging and health supplements. These areas will be the primary focus of our future work. Based on our findings, nanoprecipitation emerges as a viable method for synthesizing nano-delivery systems tailored for the food industry. Nevertheless, further research was needed to investigate the impact of QRC:β-CD NICs on cancer therapy using in-vivo models of cancer cell lines.

## Data Availability

The datasets used or analyzed during the current study are available from the corresponding author upon reasonable request.

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
