# Peer review of "Preparation and Characterization of a Nano-Inclusion Complex of Quercetin with β-Cyclodextrin and Its Potential Activity on Cancer Cells"

_micromachines, 2023, doi:10.3390/mi14071352_

Round 1
Reviewer 1 Report
The manuscript is very interesting and can be accepted for publication after addressing the following comments,
1. Some potential results should be inserted in the abstract section.
2. What are the limitations of this study.
3. 2.4. Instruments Used should be inserted in the main document.
4. Tranferr Figure 6 into Table and show the model of the statistical analysis, in one-way or two-way ANOVA.
5. Some results about the cytotoxicity and fluorescence-assisted cell sorting in the abstract section.
6. Limitation of the using Nano particles for example the toxicity.
7. Where is the ethical approval to use this material as an anticancer?
8. Deep discussion about the cytotoxicity of QRC:β-CD NICs should be inserted.
9. The mechanisms of The cytotoxicity of QRC:β-CD NICs, for example, “Concentrations up to 20.0 μg/ml showed no toxic effect on the cells, while higher concentrations of 200.0 μg/ml and 2000.0 μg/ml exhibited a less toxic effect after 3 days of treatment.
Minor corrections
Reviewer 2 Report
1. In addition to bioavailability, it would be really appreciated to highlight the facts regarding water solubility, chemical stability, absorption profile, and biochemical effects in relation to cancer.
2. The advantage of CD-based drug delivery should be pointed out briefly, emphasising its significance.
3. Provide the reference of statement “However, the majority of these techniques concentrate on chemical techniques that significantly alter the structure and properties. Additionally, several hazardous chemicals are produced during the process, which 53 may compromise the quality of food and pharmaceutical items”.
4. The title of the study is "Preparation and characterization of a nano-inclusion complex of quercetin with -cyclodextrin and its potential activity on cancer cells," but the detailed potential activity of the quercetin-cyclodextrin inclusion complex on cancer cells is lacking. It would be preferable if you addressed this under a distinct heading.
5. In the Introduction section, very old references are cited in the text while discussing dietary polyphenols and cancer. Authors can read some recent relevant references such as:
10.1016/j.neubiorev.2022.104795; 10.3389/fnut.2023.1120377; 10.1016/j.tifs.2020.10.020; 10.3389/fchem.2023.1174363; 10.3389/fchem.2023.1173624.
6. Grammatical and typo error should be corrected throughout the manuscript.
Minor Grammatical and typo errors should be corrected throughout the work.
Reviewer 3 Report
The present study authors have demonstrated the synthesis of quercetin and beta-cyclodextrin nanoparticles. The overall manuscript has been written very well and was easy to follow but lacks a few major points that need to be addressed.
Major comments:
Figure 3 for FT-IR data shows a high resemblance between the QRC alone and QRC nanocomposites. Due to the nano formulation of QRC and cyclodextrin, there should be a shift in the peaks. Please comment.
Figure 4 on NMR data should include a cyclodextrin profile as well.
In Figure 6, the authors mentioned MRC-5 cells while they talk about MCF-7 in the discussion. Please be specific. Also, there is not much effect on the cytotoxicity of MRC-5 (figure 5) even at higher dosages.
QRC in free form showed cytotoxicity on MDA-MB cells only but not on MRC-5. Though it has been reported in the literature to be cytotoxic to MRC-5 cells. Please explain.
In nanocomposite form, it showed effect only at the highest dose. If it requires such a high concentration in the 96 well-based assays. How do authors propose its advantage in patient settings?
Authors need to evaluate the release of QRC alone and QRC nanocomposites in PBS and plasma for up to 72 hours.
In brightfield imaging, QRC alone shows some cytotoxic effect on the cell types used. However, its nanocomposites do not have any effect on MRC-5 but is still toxic to MDA-MB-231 cells. This data is contradictory to the cytotoxicity data. Please explain.
Propidium iodide and annexin staining is a promising way to establish apoptotic cells. Flow cytometry data need to be re-analyzed for PE channel bleeding into FITC. It needs to be corrected before making any conclusion.
Authors claim there is an increase in apoptosis, they need to show the percentage positive cells for annexin and propidium iodide.
Minor comments:
Free QRC cytotoxicity and QRC nanocomposites should be shown on the same graph with a line graph.
English is fine.
Round 2
Reviewer 1 Report
The article has been significantly improved, thus I suggest accepting it for publication.
minor errors
Reviewer 3 Report
In Figure 6, the authors mentioned MRC-5 cells while they talk about MCF-7 in the discussion. Please be specific. Also, there is not much effect on the cytotoxicity of MRC-5 (figure 5) even at higher dosages.
Author Response : In comparison to QRC alone, the presence of high concentrations of NICs exhibits a lesser degree of toxicity in normal human lung fibroblast MRC-5 cells. Moreover, the individual and combined administration of both low and high concentrations of NICs effectively suppress the growth of cancer cells.
However, higher concentrations of 200.0 μg/ml and 2000.0 μg/ml exhibited a relatively 347 less toxic effect after a 3-day treatment period.
Figure 6 A. This is contradicting statement to the data. On day 3 there is reduced percent survival suggesting higher toxicity.
Figure 6B. QRC alone is more effective than NIC on the MDA-MB cell line. This shows QRC is alone sufficient to kill the cancer cells which goes against the advantage of formulating its nanoparticles.
QRC in free form showed cytotoxicity on MDA-MB cells only but not on MRC-5. Though it has been reported in the literature to be cytotoxic to MRC-5 cells. Please explain.
Author Response : Compared to QRC alone, the presence of high concentrations of QRC NICs inclusions demonstrates a reduced toxic effect on normal human lung fibroblast MRC-5 cells. Furthermore, both low and high concentrations of QRC NICs, whether administered separately or in combination, effectively inhibit the growth of cancer cells (MDA-MB-231).
Again, I don’t see any advantage of NIC over the free drug from the figure.
In nanocomposite form, it showed effect only at the highest dose. If it requires such a high concentration in the 96 well-based assays. How do authors propose its advantage in patient settings?
Author Response : In our study, we utilized two distinct cell lines: MRC-5, representing normal cells, and MDA-MB-231, representing cancer cells. It is important to note that our current investigation was limited to preliminary screening only. If considering the potential application of these findings in a clinical setting, further testing such as in-vivo experiments involving animal models would be necessary. These additional tests are crucial to gather more comprehensive data and evaluate the efficacy and safety of the tested approach before its potential use in human patients.
This does not answer the question that in nanocomposite form, it showed effect only at the highest dose. If it requires such a high concentration in the 96 well-based assays. How do authors propose its advantage inpatient settings?
Authors need to evaluate the release of QRC alone and QRC nanocomposites in PBS and plasma for up to 72 hours.
Author Response : Thank you for considering our scientific suggestion. Our planned research involves conducting a release study at three distinct pH levels to investigate the release behavior. Additionally, we will perform a kinetic study to further support the release mechanism. Furthermore, based on your recommendation, we will conduct the release study in plasma to better simulate the in-vitro as well as in-vivo conditions. These additional experiments will provide valuable insights into the release profile and behavior of the studied substance.
These datasets should be added to the current communicated manuscript since the authors are proposing novel NICs of QRC.
In brightfield imaging, QRC alone shows some cytotoxic effect on the cell types used. However, its nanocomposites do not have any effect on MRC-5 but is still toxic to MDA-MB-231 cells. This data is contradictory to the cytotoxicity data. Please explain.
Author Response : QRC demonstrates the ability to inhibit cell growth in both normal and cancer cells, while the nanocomposite form of QRC NICs does not impair cell growth. The incorporation of QRC into nanocomposites enhances their biocompatibility, resulting in reduced toxicity in both cell types. However, it should be noted that QRC alone does not possess the same level of biocompatibility as the fully developed QRC NICs nanocomposites.
This does not go along with your cytotoxicity data. QRC alone does not inhibit normal cell division. Please refer to the literature.
Propidium iodide and annexin staining is a promising way to establish apoptotic cells. Flow cytometry data need to be re-analyzed for PE channel bleeding into FITC. It needs to be corrected before making any conclusion.
Author Response : Apoptosis evaluation is performed using Annexin V-FITC/PI staining.
It does not answer the query raised.
Figure 7B presents the results obtained from treating cancer cells inoculated with both QRC and QRC NICs, serving as representative samples.
Authors claim there is an increase in apoptosis, they need to show the percentage positive cells for annexin and propidium iodide.
Author Response : The percentages of apoptosis observed in different conditions are as follows:
For Control : Lower Left (LL): 98.1%, Upper Right (UR): 0.5%, Lower Right (LR): 1.5%
For QRC : LL: 81.4%, UR: 7%, LR: 11.6%
For NICs : LL: 86.7%, UR: 5.7%, LR: 7.6%
This data needs to be corrected for its color compensation and then should be presented as bar graph of n=3-5 at least.